# Deep Infiltrating Endometriosis and Adenomyosis: Implications on Pregnancy and Outcome

**DOI:** 10.3390/jcm11010157

**Published:** 2021-12-29

**Authors:** Teresa Mira Gruber, Laura Ortlieb, Wolfgang Henrich, Sylvia Mechsner

**Affiliations:** 1Department of Obstetrics, Charité—Universitätsmedizin Berlin, Corporate Member of Freie Universität Berlin and Humboldt-Universität zu Berlin, Augustenburger Platz 1, 13353 Berlin, Germany; teresa-mira.gruber@charite.de (T.M.G.); laura.ortlieb@charite.de (L.O.); wolfgang.henrich@charite.de (W.H.); 2Endometriosis Centre Charité, Department of Gynecology, Charité—Universitätsmedizin Berlin, Corporate Member of Freie Universität Berlin and Humboldt-Universität zu Berlin, Augustenburger Platz 1, 13353 Berlin, Germany

**Keywords:** deep infiltrating endometriosis, adenomyosis, archimetrosis, pregnancy outcome

## Abstract

Endometriosis (EM), especially deep infiltrating endometriosis (DIE) and adenomyosis (AM), are known to cause pain and sterility in young women. More recently, they have also been described as risk factors for obstetric complications. While the pathophysiology is not yet completely understood, they seem to share a common origin: archimetrosis. Methods: A systematic literature review was conducted to summarize the existing evidence on DIE and AM as risk factors for obstetric complications. Results: Preterm birth, caesarean section delivery (CS) and placental abnormalities are associated with the diagnosis of DIE and AM. Women with AM seem to experience more often hypertensive pregnancy disorders, premature rupture of membranes and their children are born with lower birth weights than in the control groups. However, many of the studies tried to evaluate AM, EM and DIE as separate risk factors. Moreover, often they did not adjust for important confounders such as multiple pregnancies, parity, mode of conception and maternal age. Therefore, prospective studies with larger numbers of cases and appropriate adjustment for confounders are needed to explore the pathophysiology and to prove causality.

## 1. Introduction

EM is a common disease: 10% of all women between 15 and 45 years are affected [1,2]. The disease is characterized by endometrial-like tissue dispersed in other locations than the uterine cavity. One common theory of the origin of EM is the translocation of endometrial tissue by retrograde menstruation formulated by Sampson in 1927 [3]. To explain why only some women develop EM even though retrograde menstruation is a very common phenomenon, the metaplasia theory was created and adapted over the years according to new scientific evidence [4]. Typical EM lesions are located at the peritoneum of the fossa ovaricae, the rectouterine pouch, the sacro-uterine ligaments, and the apex vesicae. Lesions that infiltrate more than 5 mm into the affected tissue are considered DIE. AM is defined by endometrial-like lesions in the myometrium. There are different forms: diffuse AM with involvement of the junctional zone (JZ), and focal adenomyosis located in the outer myometrium (FOAM) often described in association with DIE. In rare cases cystic changes of the myometrium or focal adenomyomas are observed [5]. The origin of disease is not yet understood. There are concepts of metaplasia, inner and outer invasion, or invagination [6]. In line with the metaplasia theory and the concept of tissue injury and repair, Leyendecker et al. defined the concept of archimetrosis [7]. 

### 1.1. The Phenomenon Archimetrosis and Its Pathomechanisms

Magnetic resonance imaging (MRI) shows signs of AM in 79% of the women with EM [8]. This fact leads to the hypothesis that AM and EM might be manifestations of the same underlying pathology [9,10]. Leyendecker et al. describe the phenomenon as archimetrosis: uterine hyperperistalsis causes micro traumatization within the junctional zone (JZ), between endometrium and myometrium [11,12]. In adolescence and young adulthood hyperperistalsis may have a beneficial influence on conception. With time, during repetitive cycles without conception, the process becomes destructive [11]. The concept of tissue injury and repair then explains the pathophysiology of EM and AM as a vicious cycle of chronic traumatization and healing [7]. Within mechanical alteration and repetitive regeneration processes, stem cells are activated, migrate, and promote disease progression [13,14]. The involvement of stem cells also explains the fact, that EM lesions are of clonal origin [4] and appear as “miniature uteri” [15]. The associated over-expression of oxytocin and vasopressin receptors leads to structural and functional changes [16]. Pro-inflammatory mediators are released and up-regulate aromatase expression [17,18]. Increased local estrogen levels induce proliferation and angiogenesis. This mechanisms may also explain the coexistence of endometrial polyps, endometrial hyperplasia, and myomas in women with AM [19]. 

### 1.2. The Uterus as the Centre of Disease Development

During the last decades and initiated by Sampson’s transplantation theory, ectopic EM lesions were the main focus of research interest [3]. The central organ the uterus and its structural and functional changes in women with DIE and AM have been neglected. The concept of archimetrosis relocates the uterus into the centre of disease development and progression. 

### 1.3. Archimetrosis and Its Implications on Fertility

Sterility and infertility are cardinal symptoms of women with AM, EM and DIE [15]. Previous studies have described a significantly impaired sperm transport from the uterus to the tubes [20], changes in embryo implantation and an increased rate of early abortions in women with EM [7,20,21]. During the secretory phase of the menstrual cycle, progesterone plays a central role to prepare the endometrium for embryo implantation. EM is associated with changes in steroidogenesis, progesterone resistance and progesterone receptor down regulation resulting in progesterone deficiency [22]. Lower progesterone levels relatively aggravate the existing local hyper-estrogenism, attributed to the pro-inflammatory milieu [23]. Besides hormonal alterations, autoimmune processes that include activated macrophages have a negative influence on embryo implantation. Activated macrophages release nitric oxide, a free radical, that damages cell membranes and DNA with direct effect on fertility [24]. Even after successful implantation, macrophages and T-cells have the potential to attack the embryo leading to early abortions [25]. In line with these findings, women with sonographic thickening of the myometrium, as a sign of AM, show reduced embryo implantation, clinical pregnancy, and live birth rates [26]. The thickening of the uterine junctional zone can even be used as a predictor for failure in in vitro fertilization (IVF) [27]. 

### 1.4. Archimetrosis and Pregnancy

The association of EM and AM with an increased risk for the pregnant woman and the unborn child has been described before [21,28,29]. The pathophysiological mechanisms of increased infertility and miscarriage rates may also explain an increased rate of complications during pregnancy: blastocyst implantation may be compromised by uterine hyper- and dysperistalsis, leading to increased rates of placenta previa [30]. Pro-inflammatory prostaglandin-dependent processes are involved in cervical maturation, rupture of the membranes and onset of labour in the context of physiological birth. Chronic inflammatory processes in the eutopic endometrium and peritoneal space could thus increase the risk of preterm rupture of membranes (PROM) and favour preterm birth in women with EM and AM [31,32,33]. The disturbed architecture of the JZ with macroscopic thickening and an altered immunological milieu, cell proliferation and apoptotic processes could disrupt trophoblast invasion, spiral artery transformation and thus placentation at the beginning of pregnancy. In consequence, preterm birth, intrauterine growth retardation, hypertensive pregnancy disorders and premature placental abruption can arise [34]. In addition to uterine changes, extragenital EM lesions may also contribute to obstetric complications. Women with DIE suffer from intestinal adhesions and restricted mobility of the pelvic organs in relation to each other. This may lead to uterine fixation during pregnancy [30,35]. If EM-related adhesions then are put under tension by the growing uterus and erode, spontaneous hemoperitoneum may occur [36]. Moreover, EM lesions can decidually transform in pregnancy. Intestinal lesions then may perforate due to altered hormonal influences [37]. Finally, previous surgical treatment of EM or AM can also increase the risk of obstetric complications. Scarring following surgical removal of rectovaginal EM lesions can weaken the tissues of the vagina and the posterior wall of the uterus, resulting in birth injuries or uterine ruptures [37]. The risk of uterine rupture in pregnancy after uterus-preserving surgery in AM is reported to be as high as 8.7% [38].

Drawn together, women with archimetrosis develop functional and structural changes of over time. At the stage our patient seek medical treatment they may have already developed chronic pain disorders. By the time the desire to have children arises uterine function may be significantly impaired, and our patients may suffer from sterility and early abortions. If pregnancy is achieved, spontaneously, or through artificial reproductive technology, complications may arise (Figure 1).

Within the following study we aim to analyse the existing literature on the concept of archimetrosis as a risk factor for adverse pregnancy outcomes. We hypothesize that severe archimetrosis is associated with a particularly high risk for complications. Thus, we focused our search on women with AM and DIE.

## 2. Materials and Methods

The systematic literature review is based on the PRISMA statement of 2009 (see Appendix A) [39]. We searched the databases PubMed, Scopus and Web of Science. Medline and Embase were not searched separately, as these are fully contained in PubMed and Scopus, respectively. 

We identified the relevant search topics as “adenomyosis”, “deep infiltrating endometriosis”, “deep endometriosis” and “rectovaginal endometriosis”, as well as “prematurity”, “premature birth”, “cesarean section”, “mode of delivery”, “pregnancy complications”, “pregnancy outcome” and “prenatal care”. For the PubMed search, outcomes were summarised using the MeSH terms “delivery, obstetric”, “pregnancy complications” and “infant, new-born” (Search terms, see Appendix A). A search using MeSH terms is not available for Scopus and Web of Science. In order not to miss any relevant papers, Scopus and Web of Science were also searched for “obstetrics”. The search was limited to studies on humans (“humans”), articles and reviews published in journals (“Article or Review”) and texts in English, French, German or Spanish. 

The patient population (P in PICOS) was defined as women without previous surgery opening the uterine cavity (except CS) or interventional therapy for AM (e.g., radiofrequency ablation or high-intensity focused ultrasound techniques) who had given birth to at least one live or stillborn child with a birth weight above 500 g, and their new-born children. There was no criterion on mode of conception. We focused our search on patients with DIE and AM as an approximation for severe archimetrosis (I in PICOS for interventions or exposure). Studies that did not differentiate between different forms of EM were excluded. The screened results handled the classification of the individual subtypes of DIE very inconsistently. A separate inclusion criterion for the subtypes of DIE, for example according to ENZIAN [40], was therefore not formulated. Studies with patients without previous therapy for DIE and AM and those that included pre-treated patients were included. No exclusion criterion was formulated for the surgical treatment of DIE. We only considered studies that enrolled women without AM/EM as a comparator group (C in PICOS). Outcomes (O in PICOS) were type and frequency of complications during pregnancy, mode of delivery and neonatal outcome. In our research we included a wide range of study designs (S in PICOS) such as meta-analysis, reviews and observational studies with a control group that were published between 1 January 1940 and 10 March 2020. Further studies were identified by hand search and screening of the bibliographies of already included publications. 

First, the titles were screened for compliance with the inclusion criteria. Unsuitable literature was then excluded. In a second step, the abstracts were screened and publications that did not meet the inclusion criteria were discarded. Finally, the full texts were accessed. Publications whose full texts were not digitally accessible via Charité Universitätsmedizin or the Staatsbibliothek zu Berlin were not included for organizational reasons. The quality of all eligible studies was assessed using the Newcastle-Ottawa Scale (NOS) for case-control studies or the NOS for observational studies. The NOS does not define a threshold for clearly identifying studies with a risk of bias of concern. In the qualitative synthesis, only studies that were rated with at least 5 out of 9 stars were considered [41].

## 3. Results

### 3.1. Literature Search

The initial search in PubMed, Scopus und Web of Science on 10 March 2020 yielded 1.129 results. After screening according to the inclusion criteria described above and risk of bias assessment with the NOS, we included 20 studies (see Appendix A). We found three meta-analyses, two reviews, three prospective and two retrospective cohort studies and three retrospective case-control studies on obstetric complications in patients with AM, which included a total of approximately 550 patients with AM and approximately 100,000 controls without AM. For DIE, we found two reviews four retrospective case-control studies and one retrospective cohort study on obstetric complications, which included a total of 217 patients with DIE and 2336 controls without DIE (Figure 2). 

### 3.2. Preterm Birth

AM: Five cohort and case-control studies investigated the incidence of preterm birth. All showed an association between AM and preterm birth with ORs ranging from 1.96 to 5 [31,42,43,44]. We found three studies on preterm premature rupture of membranes (PPROM), as a precursor of preterm birth, in women with AM. Yamaguchi et al., Mochimaru et al. and Juang et al. showed an association between AM and PPROM with ORs ranging from 1.98 to 5.5 [31,43,45].

DIE: Three studies investigated the incidence of preterm birth. In all three studies, patients with DIE gave birth prematurely more often than controls without EM [35,45,46]. Exacoustos et al. reported a statistically significant association with an OR of 6.867 (95% CI 3.07–15.36). In the study by Mannini et al. 25% of the case-patients with DIE and only 1.2% of the controls gave birth prematurely. This difference was statistically significant [45]. Both authors did not adjust their results to important confounders. Nirgianakis et al. excluded multiple pregnancies and matched cases and controls for parity, previous CS, mode of conception and maternal age at delivery. In their study the association between DIE and preterm birth was not statistically significant [35]. No suitable studies are available on PPROM in DIE. 

### 3.3. Mode of Birth

AM: Four cohort and case-control studies investigated the association between AM and CS [42,43,44,47]. Three studies found a significant association with ORs of about 4 [42,43,47]. In the study by Shin et al. women with AM also gave birth more often via CS than women without AM [44]. However, this difference was not statistically significant. 

Hashimoto et al. described the most frequent indications for CS in women with AM as: non-reassuring fetal status (NRFS) (32.1%), abnormal fetal presentation (14.2%) and hypertension in pregnancy (10.7%) [42]. They showed a significant association between AM and CS because of NRFS with an OR of 5.07 (95% CI 1.73–14.2; *p* = 0.002) [42]. Mochimaru et al. reported abnormal fetal presentation (25.0%), NRFS (22.2%) and previous CS (8.3%) as the most common indications [47]. 

In two studies, abnormal fetal presentation was described as more common in pregnancies in women with AM than in controls [42,47]. 

Three studies addressed peripartum hemorrhages in women with AM and reached different conclusions: Mochimaru et al. looked at singleton pregnancies in 36 women with and 144 controls. They calculated an OR of 5.0 (95% CI 2.2–11.4) and adjusted for maternal age [47]. Hashimoto et al. studied peripartum hemorrhages separately in spontaneous deliveries and CS, respectively. In both CS and spontaneous deliveries, 49 women with AM experienced a higher blood loss than 245 controls. These differences were not statistically significant. They excluded multiple pregnancies and matched for parity, mode of conception and maternal age [42]. Sharma et al. compared pregnancies after IVF/ICSI in 22 women with AM and 140 controls. In both groups, peripartum hemorrhage occurred in about 9% of women [48]. 

DIE: In two Italian studies by Uccella et al. and Exacoustos et al. women with DIE gave birth by CS significantly more often than the controls [46,49]. Exacoustos et al. reported an OR of 2.817 (95% CI 1.404–5.651) [46]. Nirgianakis et al. addressed indications for CS in women with DIE: 32.2% of patients delivered by primary CS and 25.8% by secondary CS. The most common indications for primary CS were breech presentation and previous CS section, each with six out of 20 cases. The most frequent indications for a secondary CS were labour dystocia and pathological CTG, each with six out of 16 cases. One woman with DIE underwent a CS because of a spontaneous hemoperitoneum. During the operation, the team discovered an intra-abdominal hemorrhage from an EM site [35]. Nirgianakis et al. studied 26 women with DIE who successfully gave birth vaginally. Prior to pregnancy, 14 women underwent resection of the vaginal fornix. Four had a bowel resection and eight underwent bowel shaving. They grouped complications of vaginal delivery into three mixed endpoints: perineal tear II° or episiotomy; perineal tear III° or buttonhole tear IV°; vaginal tear. No significant differences were found between women with DIE and women without EM for any of the three endpoints, and none of the cases or controls experienced a severe vaginal tear [35]. In line with these findings in the study of Exacoustos et al. woman with DIE had no more severe vaginal tears, cervical tears or atonic bleeding than the controls. They also analysed CS associated complications. Women with DIE experienced significantly more often a hysterectomy, a hemoperitoneum or bladder injuries (for each of the three complications: 2/41 cases vs. 0/300 controls; OR = 24.62; 95% CI 1.149–527.7) [46]. One women with DIE required resection of part of the bowel during CS. Three of the included studies investigated peripartum hemorrhages in women with DIE compared to women without EM. The three studies showed no common trend and none found a statistically significant difference [35,45,49]. Among the included studies, only Mannini et al. collected data on induction of labour in DIE and found no significant difference between women with and without DIE [45].

### 3.4. Other Complications

AM: Placental malposition, gestational hypertension, preeclampsia (PE) and PPROM as risk factors for preterm birth were more prevalent in pregnant women with AM than in controls. Two studies found significantly higher rates of placental malposition in women with AM [42,50]. Hashimoto et al. reported an OR of 4.9 (95% CI: 1.4–16.3) [42]. Five studies investigated hypertensive pregnancy disorders in patients with AM. Porpora et al. and Hashimoto et al. found significantly increased rates of gestational hypertension and PE in women with AM [42,51]. Harada et al. reported an OR of 1.86 (95% CI: 1.11–3.14). In the studies by Sharma et al. and Mochimaru et al., women with AM were also more likely to develop PE than controls without AM [47,48]. 

The new-born infants of pregnant women with AM in the included studies were significantly more often lighter than 2500 g and 1500 g, and more often below the 10th weight percentile than the new-born infants of controls [42,43,44,47]. Umbilical artery pH, APGAR at five minutes and neonatal intensive care unit admission rates of infants born to mothers with AM were not significantly different from controls [42,47]. 

DIE: Three studies comparing pre-operated women with DIE with women without EM showed a significant association between DIE and placenta previa [35,46,49]. 

In three observational studies gestational hypertension was significantly more common in pregnant women with DIE than in in controls [35,45,49]. For PE, the available studies found no significant association with DIE. 

Two studies addressed premature placental abruption in patients with DIE. Exacoustos et al. calculated an OR of 15.33 (95% CI 1.359–173). They included 41 case-patients with a persistent DIE nodule of more than two centimetres and did not adjust for confounders [46]. In contrast, Nirgianakis et al. compared singleton pregnancies in 61 women with posterior DIE with 186 controls. Cases and controls were matched for previous CS, parity, mode of conception and maternal age. They found no significant difference in premature placental abruption between the two groups [35]. 

The literature does not show a significant association with DIE for any of the neonatal outcome endpoints investigated.

## 4. Discussion

Our analysis revealed that women with DIE and AM seem to have an increased risk for preterm birth, CS, placental malposition and insertion disorders, placental abruption, gestational hypertension, and PE, even though not all the associations described are significant. 

### 4.1. Archimetrosis and Its Implications for the Mode of Birth

The CS rate in all analysed studies was higher in women with DIE and AM than in the control groups, even though they differed widely between the studies. In women with DIE the rates ranged from 32.3% found by Nirgianakis et al., to 68.3% found by Exacoustos et al. [35,46]. The CS rates of the control groups also differed significantly, which makes a comparison of the results difficult (21%, Nirgianakis et al. vs. 43.3%, Exacoustos et al.). The included studies handled important confounding factors such as parity, previous CS, mode of conception and multiple pregnancies differently. We assume this is the main reason for the marked heterogeneity of the results. For example, in the case-control study by Exacoustos et al. the proportion of pregnancies after assisted reproduction in the case group was 43.9%, while only spontaneously conceived pregnancies were included in the control group. In addition, age and parity of cases and controls differed significantly. Uccella et al. and Mannini et al. did not adjust for maternal age and assisted reproductive technology (ART) and found increased CS rates in women with DIE [45,49]. 

Moreover, CS rates are dependent on a variety of socio-cultural and structural factors. These would be another explanation for the fact that CS rates in women with DIE and AM greatly differ in different studies—depending on the time and place of patient recruitment and the type of hospital. There is also a difference in CS rates among countries, with, for example, Italy having the highest CS rate reported in Europe lately (up to 60% in certain regions) [52]. In this context, the high CS rates in the study of Exacoustos et al. may be interpreted as country specific. A direct comparison of CS rates among different studies seems not expedient.

It is important to keep in mind that in the study by Exacoustos et al. there were significantly more hysterectomies, haemoperitoneum and bladder injuries during the performance of CS in women with DIE compared to women without EM [46]. At the same time, there were no more complications in vaginal deliveries in women with DIE than in women without EM [35,46]. A French observational study by Thomin et al. examined 72 pregnancies in women with DIE, without a control group. It described significantly more complications in deliveries by CS than in vaginal deliveries. In particular, anterior DIE was associated with difficulties in child extraction during CS [53]. Moreover, having a CS seems to be the most common cause of scar EM [54]. 

In conclusion, the indication for a CS in women with archimetrosis should be evaluated carefully to avoid severe complications.

### 4.2. Underlying Causes of Preterm Birth

We found that women with archimetrosis have an increased risk to give birth prematurely [31,42,43,44,45]. The underlying causes of birth promoting inflammatory processes and changes within the JZ are not yet clear [31,32,33,55] Despite this, a controversial statement was made by Leone et al.; they suggested that the increased rate of preterm birth in women DIE may not only be due to the disease itself, but rather iatrogenic, for example, triggered by the concept of “the precious baby” [37]. “An intriguing challenge of future studies will be to distinguish between spontaneous and iatrogenic preterm birth, answering the question of whether EM per se may cause uterine contractions and preterm labour or may create clinical or behavioural conditions (bleeding, abdominal pain, anxiety in the operators) that can result in a CS in early gestational ages” [37]. 

### 4.3. Placental Abnormalities and Its Association with Pregnancy Complications

In our analysis, increased rates of placenta previa in women with DIE and AM were described [35,42,45,46,49,50]. Moreover, studies showed a significant association between DIE and hypertensive pregnancy disorders [35,49], and gestational hypertension and PE occur significantly more frequently in women with AM than in controls [36,42,47,48,50,51,56]. Despite the influence on the mode of delivery, placental problems such as placental insufficiency are associated with intrauterine growth restrictions. It seems therefore not surprising that women with AM have an increased risk of giving birth to infants below the 10th weight percentile and lighter than 2500 g and 1500 g, respectively [42,43,44,47]. 

In this context, the changes of the JZ and the impact on spiral artery development seem to be a topic of future research interest. 

## 5. Limitations

### 5.1. Retrospective Study Design

Most of the studies followed a retrospective study design [31,35,42,44,45,46,47,48,49,57]. A retrospective study design is associated with a particular risk of misattribution both at the level of exposure and at the outcome level, because either existing medical documentation or retrospective surveys are used as a source of information. The purpose of documentation in everyday medical documentation is not science but the pragmatic care for women which affects the validity of the documented data. Retrospective surveys are particularly at the risk of recall bias. Moreover, the studies on DIE included only a small number of cases. 

### 5.2. Misclassification

The AM studies are at high risk of misclassification bias. It is difficult to find a generally accepted consensus on valid radiological criteria for the diagnosis of AM. In addition, only four of the included studies classified the cases according to defined criteria prior to conception [31,47,48,51]. Two studies performed ultrasound diagnostics as part of prenatal care in the first trimester, although there are no validated ultrasound criteria for AM in pregnancy [42,44]. Two studies referred to the Japan Environment and Children’s Study, a national Japanese birth cohort study [43,50]. Classification of cases and controls for these two studies was based on woman’s statement of AM status in a questionnaire. The classification of case-patients in most of the included studies on DIE is more precise, as the most of the study authors referred to histological examinations after surgical treatment for EM [35,45,46,49]. Another risk for misclassification bias is the difference in diagnostic methods for cases and controls in the studies. The diagnosis of DIE was often based on laparoscopy and histology, while controls were only asked about typical symptoms [35,45,46,49].

### 5.3. Selection Bias

Especially in the included case-control studies, but also in some of the cohort studies, the selection of controls carries a high risk of selection bias. Most studies either drew their controls from small specific groups, such as IVF pregnancies in women with tubal infertility [48], or looked at women giving birth in university hospitals and other maximum care hospitals [44,51]. Since women with special risk factors or complications in pregnancy are usually advised to give birth in a maximum care hospital, a higher rate of reported complications in pregnancy, more preterm births, higher CS rates and a worse neonatal outcome are expected. This probably leads to a bias towards the null hypothesis, and significant effects that were observed do not seem less credible. 

### 5.4. Confounding

The included studies dealt differently with possible confounders such as parity, previous CSs, multiple pregnancies, mode of conception and maternal age. Each considered a variety of outcomes and conducted statistical tests for all of these outcomes without adjusting for multiple testing. This results in the potential for randomly significant results in almost all included studies. 

### 5.5. Heterogenicity

In addition to the quality of the individual papers, a major problem in the work on this review was the great heterogeneity of the study designs. In particular, the studies on DIE can only be compared to a limited extent. In part, pre-operated women were included after complete excision of all DIE foci and in part women who were not operated at all, or women in whom EM foci were still detectable sonographically after surgery. The surgical methods also differed. A comparison of the studies is also complicated by the fact that the background risks for the individual outcomes seem to vary depending on the setting. 

## 6. Conclusions

We recommend women with archimetrosis to give birth in an experienced obstetric setting. In antenatal care, placental insertion disorders need to be ruled out by targeted ultrasound examinations. Ideally, there should be the possibility of consulting an EM specialist with surgical experience. In addition, the indications, especially for primary CS, should be evaluated carefully. Future basic research as well as prospective studies with appropriate case numbers and wise adjustment for relevant confounders such as parity, previous CS, multiple pregnancies, previous illnesses, maternal age and mode of conception are necessary to verify our results.

## Figures and Tables

**Figure 1 jcm-11-00157-f001:**
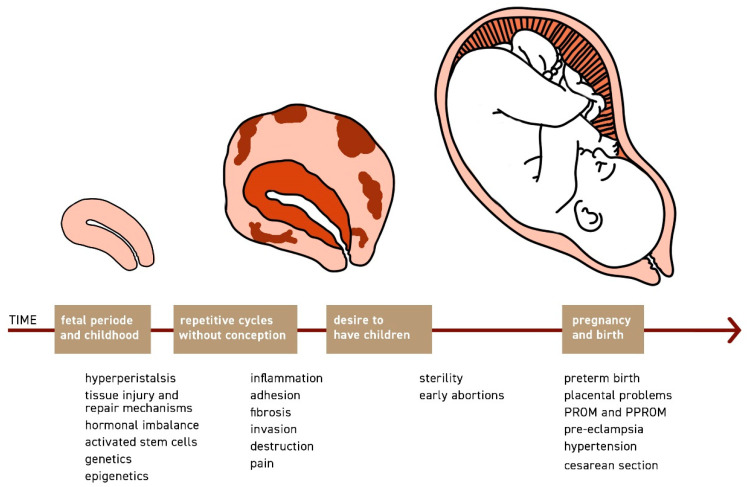
Women with archimetrosis develop functional and structural changes of the uterus over time. When the desire to have children arises the uterine function may be impaired and symptoms such as sterility and early abortions are commonly observed. Even if pregnancy is achieved spontaneously, or more likely with the help of IVF, it may be associated with complications such as preterm birth, placental problems, early rupture of membranes, pre-eclampsia, hypertension, and a higher rate of CS.

**Figure 2 jcm-11-00157-f002:**
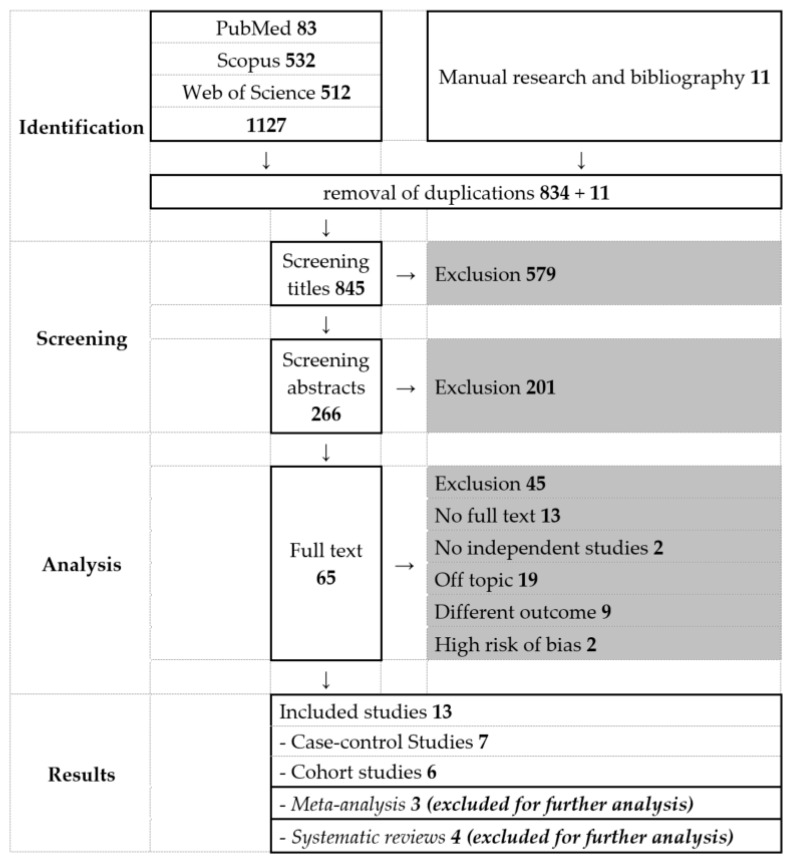
PRISMA diagram.

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
