# Peer review of "Deep Infiltrating Endometriosis and Adenomyosis: Implications on Pregnancy and Outcome"

_jcm, 2021, doi:10.3390/jcm11010157_

Round 1
Reviewer 1 Report
Deep infiltrating Endometriosis and Adenomyosis: implications on pregnancy and outcome
A systematic literature review summarising the existing evidence on adenomyosis and DIE as risk factors for obstetric complications as preterm birth, cesarean section delivery and placental abnormalities.
I am surprised by the small number of studies on this topic (13) that you obtained using the PRISMA statement 2009.
Otherwise, it is certainly a current topic that will interest the reader.
1, I see a weakness in your work in the poorly defined difference between Endometriosis and DIE. In practice, the article erases whether these are peritoneal endomeriosis or endometriomas or deep infiltrating endomeriosis. This results in an incredibly heterogeneous group
2, There is a lack of a clear clinical or operational assessment of DIE, using ENZIAN or r ASRM or Fertility index. This makes it difficult to evaluate the final results of the studies.
3, Similarly, the evaluation of adenomyosis is missing - either by MR or ultrasound.
I don't know why Figure 1 is shown . There is no description of the picture to evaluate the severity and possible types of adenomyosis. An ultrasound finding might be more appropriate.
Author Response
Please find my answer attached

Reviewer 2 Report
Review Manuscript ID: jcm-1482735
Deep infiltrating Endometriosis and Adenomyosis: implications on pregnancy and outcome
Interesting and thorough work.
Introduction
I find that the introduction is very long and does not read well. Either shorten it and cite the papers later on or divide into several paragraphs.
The MRT image is nice but I suggest also to add an ultrasound image, or skip both.
Line 23 – “interfere causality” not clear. I think a word is missing
Line 68 – the word been is written twice
Materials and methods
Please separate into several paragraphs.
Results
You should make sure that abbreviations are explained. For example, line 155, PPROM and there are more
Line 167 – the word significant appears twice
Make sure you go over the entire manuscript with a speller or give it for language editing
Discussion
I didn’t quite follow your explanation in lines 263-267. Most complications are probably resulting from adenomyosis. We do know that there is an association between DIE and adenomyosis and this likely the reason for the association with DIE. There is also an association between endometriosis overall (also low-grade versions) and adenomyosis and this needs to be discussed and possibly referenced.
Line 282 – It is worthwhile to find out what the baseline CS rates are in these countries. This is likely the reason for the differences.
Line 305 – not clear whether you are referring to AM or EM
Paragraphs please……….
Conclusion
There is no need for lines 348 – 361. In fact, the conclusion starts with line 362.
Consider moving lines 348-361 and the associated figure elsewhere.
Author Response
Please find my answer attached
